# The Effect of Shot Peening on Residual Stress and Surface Roughness of AMS 5504 Stainless Steel Joints Welded Using the TIG Method

**DOI:** 10.3390/ma15248835

**Published:** 2022-12-10

**Authors:** Magdalena Bucior, Rafał Kluz, Tomasz Trzepieciński, Kamil Jurczak, Andrzej Kubit, Kamil Ochał

**Affiliations:** 1Department of Manufacturing and Production Engineering, Rzeszow University of Technology, al. Powst. Warszawy 8, 35-959 Rzeszów, Poland; 2Mechanical and Electrical Engineering Department, Polish Naval Academy, 81-103 Gdynia, Poland; 3Department of Materials Science, Rzeszow University of Technology, al. Powst. Warszawy 8, 35-959 Rzeszów, Poland

**Keywords:** shot peening, stainless steel, surface roughness, TIG (Tungsten Inert Gas)

## Abstract

This article presents the influence of the Shot Peening (SP) process on residual stress and surface roughness of AMS 5504 joints welded using the Tungsten Inert Gas (TIG) method. Thin-walled steel structures are widely used in the aviation and automotive industries, among others. Unfortunately, the fatigue properties become worse during the welding process. Samples of 1 mm-thick AMS 5504 steel plates were first prepared using TIG welding and then strengthened by the Shot Peening (SP) process. The technological parameters of the SP process were changed in the range of time t from 2 min to 4 min and of pressure *p* from 0.4 MPa to 0.6 Mpa. The residual stresses were measured by X-ray diffraction in three zones: fusion zone (FZ), heat-affected zone (HAZ) and base metal (BM). The results showed that SP introduced compressive residual stresses in all of the zones measured, especially in the FZ. The greatest value of compressive residual stresses σ = −609 MPa in the FZ was observed for the maximum parameters of SP (*p* = 0.6 MPa, t = 4 min). The increase in value of residual stress is about 580% when compared to welding specimens without treatment. As a result of shot peening in the FZ, the mean roughness value Ra decreased in range 63.07% to 77.67% in the FZ, while in the BM increased in range 236.87% to 352.78% in comparison to specimen without treatment. Selected surface roughness parameters in FZ and BM were analyzed using neural networks. In FZ, it was demonstrated that the most correlated parameters with residual stresses are Rt and Rsk. On the other hand, in the BM zone, the most correlated parameters were Rv, Rt and Rq. This enables the estimation of stresses in the welded joint after SP on the basis of selected roughness parameters.

## 1. Introduction

Thin-walled welded structures are widely used in some components for the aviation and automotive industries. Combining them is a very important issue in today’s manufacturing industry. This process should be safe, cost-effective and should ensure high quality of the connections obtained. Stainless steel elements can be joined using various techniques. The use of friction welding makes it possible to reduce the cost of the process, due to the fact that it does not require the use of shielding gases and special holders [1,2]. However, welding is the most commonly used method for joining thin-walled structures. Obtaining the required joint quality often requires process optimization [3]. However, despite this, this process introduces tensile stresses in the surface layer of the joint, which reduce its fatigue strength. Taking into account the thermal influence of the welding process, fatigue-weak areas exist, especially in the fusion zone (FZ) and heat affected zone (HAZ) of welded structures [4]. One of the ways of improving the strength properties in these zones is the surface strengthening treatment after welding, such as shot peening (SP) [5,6], laser shock peening (LSP) [7], ultrasonic peening [8], water jet peening (WJP) [9] and fine particle peening technology [10]. During the SP process, the surface is bombarded by small spherical balls. This treatment introduces beneficial residual stresses thereby increasing the fatigue strength. Torres et al. [11] demonstrated that the fatigue strength of the base plate treated by SP was greatly enhanced due to compressive residual stress induced not only in the steel surface but also down to a certain depth. Similar results were found by Kinoshita et al. [12]. They found that SP introduced compressive residual stress in the welded joints not only at the surface but also at least up to a depth of 400 μm. Mori and Ogawa [13] and Yamada et al. [14] reported that the fatigue strength improvement of the SP welded joints was expected in stress ratio, R = –1 or R = 0 and certain compressive residual stress was confirmed at the weld toe. Shi et al. [7] investigated the effect of LSP on the fatigue properties of 3 mm-thick welded Ti-6Al-4V titanium alloy plate. The results indicated that the fatigue strength was enhanced by 19.82% after LSP. Furthermore, the Vicker’s hardness increased in the FZ and HAZ by 8.56% and by 3.79%, respectively. They also indicated that the LSP process changed tensile residual stress into compressive residual stress. In turn, Chandrasekar et al. [15] examined the non-peened and LSP dissimilar welds of Inconel 600 and AISI 316L produced by the Tungsten Inert Gas (TIG) welding technique. It was found that before LSP treatment, the tensile fracture occurred in the weld zone and the tensile strength was lower than that of the base metal. After LSP treatment, the tensile fracture occurred far away from the weld and it was on the side of the weaker metal AISI 316L. The authors also stated that during the LSP process, the tensile residual stresses of non-peened welds were transformed to compressive stresses and microhardness increased after peening, especially in the FZ and HAZ. Strivastava et al. [9] studied the improvement of residual stress and subsurface hardness of stainless steel using pulsed water jet peening (PWJP). They analyzed welded AISI 304 stainless steel with a thickness of 5 mm. It was observed that maximum residual stress increased from −67 MPa to −332 MPa in the FZ and from −122 MPa to −449 MPa in the HAZ. The surface hardness of 3.12 GPa increased up to a maximum of 5.43 GPa after PWJP. Moreover, the arithmetical mean deviation of the profile (mean roughness) increased from Ra = 1.84 µm to Ra = 3.47 µm at a pressure of 40 MPa.

SP can refine the grain size of material and this effect depends on the peening intensity [16,17] and coverage [16,18]. Jamalian and Field [19], Maleki and Unal [16] and Zhang et al. [20] found that the finest grain usually appears on the surface or subsurface, and the grain size gradually increases deeper in the material. When plastic deformation occurs, the dislocation density increases, resulting in the formation of dense dislocation walls and dislocation tangles [21,22].

An important issue discussed in the publications is also the analysis of the geometrical structure of the surface because it also has a significant impact on the fatigue strength and both wear and corrosion resistances. In many publications, when authors analyze the surface roughness after shot peening, they pay attention to one parameter, Ra. Surface characteristics based on only one parameter are insufficient. Other height and amplitude parameters should also be analyzed, such as Rt, Rz, Rq, Rsk or Rku. Grzesik [23] presented the relationship between the surface roughness parameters and functional properties of machine parts after different machining processes. He pointed out that fatigue strength is significantly influenced by the Rt and Rz parameters, and to a lesser extent by the Ra, Rq and Rku parameters. Another important issue is the correlation of the parameters of surface roughness with the functional features of the surface, so that the measurements can be used to predict specific functional properties, e.g., fatigue strength. This would significantly shorten the time needed for research. Lin et al. [24] and Liu et al. [25] found that the roughness parameter Sa increases with the growth of SP velocity. According to Maleki et al. [26] the use of double SP can only slightly decrease the roughness parameter Sa. Singh et al. [27] found that double SP reduces the surface roughness without significant change in residual stress. As the coverage increases from 100% to 400%, surface roughness parameters Sa, Sq, S5z, Sku decrease [28].

Many researchers focus on the effect of SP on welded joints, and investigate the welding of various materials. None of them have focused on the selection of parameters for this process, which are especially important for thin-wall welded joints. The aim of the work was to select the technological parameters of the SP process for welded joints made of AMS 5504 stainless steel with a thickness of 1 mm. Hence, this research has been carried out on the influence of various SP parameters on the residual stress distribution and surface roughness in three zones of the weld (FZ, HAZ and BM).

A novelty in the article was the use of artificial neural networks to demonstrate the relationship between compressive stresses and selected roughness parameters of welded joints after SP. This makes it possible to determine the stress values of the joints made only on the basis of the knowledge of roughness parameters, which are much easier to measure than stresses. Roughness measurement is less time-consuming and much less expensive, and does not require specialized equipment.

## 2. Materials and Methods

### 2.1. Test Material

The welding experiments were performed using AMS 5504 (410) martensitic stainless steel sheet metal with a thickness of 1 mm. The chemical composition and mechanical properties of AMS 5504 steel are shown in Table 1 and Table 2, respectively. This grade of steel is commonly used in many industries such as aviation (for elements of aircraft structures), automotive (collectors and high-temperature engine components) and in medicine for medical instruments.

### 2.2. Welding Procedure

Two strips with dimensions 400 mm × 100 mm × 1 mm were welded using TIG welding technology. The parameters of the welding process were selected on the basis of preliminary tests performed for various variants. The parameters ensuring the highest quality of the weld were selected for the tests. The welding parameters are shown in Table 3. Next, ten specimens were cut from the welded sheet. Nine of these were shot peened and one specimen was left as a reference. The values of two technological parameters of SP were set at three levels, time t in the range from 2 to 4 min and pressure p in the range from 0.4 MPa to 0.6 MPa (Table 4). The diameter of the ball bearings, which was d = 1.5 mm, was a constant parameter in the tests. The distance of the specimen from the nozzle was 250 mm.

### 2.3. X-ray Diffraction Analysis

A Proto iXRD Combo instrument was used to measure the residual stress by the X-ray diffraction method. A lamp with a chrome anode and a beam radiation Cr Kα with a wavelength λ = 2.291Å was used in the experiments. The voltage and current were 20 kV and 4 mA, respectively. The measured values of the interference peaks were evaluated by the sin2ψ [30] method with the diffraction angle (2θ) varying between −25° and 25°. The residual stresses were measured in three zones: FZ, HAZ and BM. The measurements were carried out in a transverse direction to the weld (Figure 1).

Figure 2 shows the micrographs of individual zones showing the characteristics of their structure. FZ is characterized by the grain pattern formed as a result of the nucleation and growth of recrystallized grains. Additionally, one can observe recrystallization and grain growth in HAZ caused by temperature increase and the columnar dendritic formation in the transition zone at the border of FZ and HAZ.

### 2.4. Surface Roughness Analysis

The surface roughness measurements were carried out according to EN ISO 4287:1999 [31] using the Talysurf CCI Lite optical profilometer. The amplitude parameters selected were analyzed in two zones: FZ and BM with triple repetition. Next, the results obtained were subjected to statistical analysis. The conditions for conducting the experiments ensured the repeatability of the results.

### 2.5. Artificial Neural Networks

The choice of roughness parameters that characterize the quality of the treated surface, which in turn is correlated with the value of residual stresses, is an extremely difficult task if an analytical solution is to be obtained. For this reason, artificial neural networks (ANNs) were used, which, as a result of a training process, made it possible to find the rank of individual roughness parameters in the impact on the residual stresses. The Statistica program was used for the investigations. Multilayer networks with one hidden layer were analyzed (Figure 3) since these are capable of modelling functions of any complexity [32]. The roughness parameters Ra, Rku, Rq, Rsk, Rt, Rv and Rz were applied as input parameters. Residual stress was considered as the dependent (output) variable. The significance of the influence of the roughness parameters on the value of the residual stresses was considered separately for the BM and FZ. The back propagation algorithm, which is the most frequently used algorithm for training multilayer networks [33], was applied to train the ANNs. The value of the coefficient of determination R2 and standard deviation (SD) ratio were adopted as the criteria of network quality.

## 3. Results and Discussion

### 3.1. Cochran Test

In the first stage of the research, the repeatability of the experimental conditions was checked. The measurements of residual stresses in three zones and roughness measurements in two zones were carried out with triple random repeatability. The repeatability of the experimental conditions was assessed based on the Cochran criterion [34]. This involved the determination of the G value according to Equation (1), which was then compared with the critical value G_kr(α;f1;f2)_ with the adopted significance level α = 0.05. In the investigations, the number of degrees of freedom for the numerator was f_1_ = 9, and for the denominator f_2_ = 2. For the measurements examined, the critical value was G_kr(0.05;9;2)_ = 0.47. The last step was to compare the G value obtained with G_kr_. Condition (2) was satisfied, therefore the experiments were carried out in a statistically significant manner.
(1)$$G=\frac{S^{2}{\left(y\right)}_{imax}}{\sum_{i=1}^{N}S^{2}{\left(y\right)}_{i}}$$
where: $S^{2}\left(y\right)$ is the variation in measured values; N is the total number of experiments.
(2)$$G<G_{kr}$$

From the analysis of the results obtained from the calculations of the G factor (Table 5) for the roughness parameters and residual stresses, the repeatability of the experimental conditions can be considered satisfactory.

### 3.2. Surface Roughness

Investigations of the surface roughness of the fusion zone (FZ) have shown that SP has a positive effect on the properties of the surface layer. The topographic maps presented in Figure 4a and Figure 5a show that shot peening process in FZ causes a significant reduction in the unevenness of the face formed during welding. In turn, the view presented in Figure 5b shows the numerous dimples typical of the peening process.

The analysis of the height and amplitude parameters shows that the SP significantly reduces the values of the parameters examined in the FZ compared to the non-peened specimens (Table 6). In turn, it can be observed that the values of most of the parameters examined in the BM (Table 7) increased in addition to the amplitude parameters Rku and Rsk.

The kurtosis Rku, which is a measure of the slope of the amplitude density curve of ordinate values of the roughness profile after peening, decreased in both zones FZ and BM compared to the base variant (the non-peened specimen). Values of Rku < 3 indicate that the shape of the amplitude density curve is flat. This means that shot peening softened the sharp peaks and grooves when compared to the surface after welding where Rku = 3.82 in the FZ and Rku = 4.21 in the BM. Skewness Rsk, which is a measure of the asymmetry of the amplitude density curve in the FZ after treatment, is negative and is within the range from Rsk = −0.47 to Rsk = −0.15. For the base variant, skewness is positive with Rsk = 0.86. Thus, it can be concluded that the shot peening of the surface of the welded joint flattened the elevations and rounded their peaks. The skewness (Rsk < 0) and kurtosis (Rku < 3) obtained after peening indicate a favorable profile concentration along the line. Taking into account the considerations regarding the performance of welded joints in use, it can be said that shot peening has a positive effect on the state of the surface layer.

The arithmetical mean deviation of the profile measured in FZ on the non-peened specimen is Ra = 4.0 μm and decreases as a result of SP to Ra = 0.89 to 1.47 μm (Table 6). In turn, in BM the Ra parameter increases from Ra = 0.37 μm (base variant) to Ra = 1.27 to 1.70 μm (Table 7). The lowest values of the Ra parameter were obtained in the FZ shot peened at a pressure in the range *p* = 0.5 to 0.6 MPa. Similar relations can also be observed for the amplitude parameter Rq. The value of the root mean square deviation in the FZ decreases from Rq = 5.04 μm to Rq = 1.13 to 1.84 μm after peening treatment (Table 6).

The analysis of the changes in the maximum profile valley depth Rv, maximum height Rz and total height Rt shows that shot peening in the FZ had a beneficial effect on the state of the surface layer, reducing the irregularity of surface created during welding. The smallest values of the height parameters were obtained for the maximum shot peening parameters, i.e., *p* = 0.6 MPa, t = 4 min (Table 6). The Rv, Rz and Rt parameters measured on the SP surface decreased by 93%, 91% and 69%, respectively. The smallest values of the parameters Rv = 2.91 µm, Rz = 6.19 µm and Rt = 8.81 µm in BM were obtained for variant 7, where *p* = 0.4 MPa and t = 4 min. Both in FZ and BM, the SP at pressure *p* = 0.4 MPa and t = 2 min (variant 9) did not reduce the profile height. Shot peening at t = 4 min (variant 1, 4, 7) brought the best results, causing a reduction in surface roughness in the FZ and the smallest increase in surface roughness in the BM.

### 3.3. Residual Stress

The results of residual stress measurements are presented in Figure 6. The results show that SP introduces significant compressive stresses in all the zones tested. The highest compressive stresses can be observed after SP in the FZ ranging from −354 MPa (variant 9) to −609 MPa (variant 1), which is an increase of 379% and about 580%, respectively in comparison with the base variant (σ = 127 MPa). The values of compressive stresses in the HAZ and BM after SP are comparable and are within the limits of measurement error. It can be observed that with the increase in SP parameters, the value of compressive stresses increases. The greatest values of stresses were obtained for the SP time t = 4 min and the pressure *p* = 0.6 MPa. These values in the FZ, HAZ and BM are σ = −609 MPa, σ = −358 MPa and σ = −342 MPa, respectively. On the other hand, the lowest value for variant 9 (*p* = 0.4 MPa, t = 2 min) was not different in the FZ, HAZ and BM, σ = −354 MPa, σ = −218 MPa and σ = −207 MPa, respectively.

The plastic deformation occurring in the surface layer under the influence of the SP process can be considered as a process of generating new dislocations and moving them in the crystallites. An increase in the number of defects in the lattice in the deformation process causes a decrease in the density of the metal (i.e., an increase in its specific volume). Since the deeper, undeformed layers do not change, compressive stresses are constituted in the surface layer, which later transform into tensile stresses that balance them.

Compressive stresses introduced in the joint area are beneficial from the point of view of fatigue life. The introduced state of compressive stresses in the surface layer of the joint contributes to the delay in the initiation of fatigue cracking and, in the initial phase, limits the propagation of fatigue cracks.

### 3.4. Sensitivity Analysis

The relationship between the roughness parameters and residual stress was determined separately for the BM and FZ zones. ANN analyses were performed with a different number of neurons in the hidden layer in order to obtain the highest value of the coefficient of determination R^2^ and at the same time the lowest value of the SD ratio. The neural networks which ensure the fulfilment of the above-mentioned conditions for the BM and FZ have the following structure: 7:7–7-1:1 (seven neurons in the hidden layer) and 7:7–9-1:1 (nine neurons in the hidden layer), respectively. The R^2^-value for the network after the training process was above 0.94 (Table 8). A better value of the SD ratio (0.17597) was obtained for the ANN modelling the residual stress in the BM. The smaller the value of the SD ratio, the better the model’s ability to generalize the data. For a very good model, the SD ratio is less than or equal to 0.1.

A sensitivity analysis (Table 9 and Table 10) was performed to determine the correlation of individual roughness parameters with the value of the residual stresses. The total network error after removing the variable corresponding to the given column from the input data set is shown in the ‘Error’ row. The error value obtained in this way can be compared with the error of the network having a full set of input variables. It is assumed that a given explanatory variable is the more important the greater the increase in the error value caused by its removal. The ‘Rank’ row contains the ordinal numbers of the variables ordered according to their importance. The ‘Ratio’ row contains the ratio of the error obtained after removing the selected explanatory variable and the error obtained with the help of a network which contains all the explanatory variables. Rt is the most strongly correlated parameter of surface roughness in relation to the calculated values of residual stresses in both the FZ (Table 9) and BM (Table 10). The Rz and Rku parameters show the lowest information capacity.

### 3.5. Regression Analysis

For the most strongly correlated surface roughness parameters with residual stresses, an adequate mathematical model was determined in the form of a regression function *W*(*x*). Regression analysis was carried out using the least squares method with the following criterion assessing the quality of the approximation [35]:(3)$$\min R=\min\overset{N}{\underset{i=0}{\sum }}{\left[f\left(x_{i}\right)-W\left(x_{i}\right)\right]}^{2}$$
where the value of the *R* function is a certain measure of the deviation of the approximation function *W*(*x*) from the approximated function *f* (*x_i_*), *i* = 1, …, *N*; *N* is number of experiments.

During the approximation, the most frequently selected basic functions are monomials according to Weierstrass’s theorem. This theorem defines that for every function *f*(*x*) specified and continuous on a closed and limited interval [*a*, *b*] there exists a polynomial *W = b_0_ + b_1x1_ + b_mxm_*, that approximates monotonously the function *f*(*x*) on the interval [*a*, *b*]. During the analysis, however, it was not possible to obtain such a polynomial with a rational m level which could be considered adequate. Therefore, the m-degree algebraic polynomial was adopted for the definition of the interactions be-tween the roughness parameters:(4)$$W\left(x\right)=b_{0}+\sum_{i=1}^{S}b_{i}^{\left(1\right)}x_{i}+\sum_{\begin{array}{l} i,j=1\\ i<j \end{array}}^{S}b_{ij}^{\left(1\right)}x_{i}x_{i}+\sum_{\begin{array}{l} i,j,\ldots ,l,n=1\\ i<j,\ldots ,l<n \end{array}}^{S}b_{ij\ldots \ln}^{\left(1\right)}x_{i}x_{i}\ldots x_{l}x_{n}+\sum_{\begin{array}{l} i,j=1\\ i\neq j \end{array}}^{S}b_{ij}^{\left(2\right)}x_{i}^{2}x_{j}+\sum_{i=1}^{S}b_{ii\ldots m}^{\left(m\right)}x_{i}^{m}$$
where $b_{0},\;b_{i}^{\left(1\right)},\;b_{ii}^{\left(1\right)},\;b_{ij\ldots ln}^{\left(1\right)},\;b_{ij}^{\left(2\right)},\;b_{ii\ldots m}^{\left(m\right)}$ are unknown coefficients, while i, j,…, *n* = 1,…, S variables of a polynomial (4).

The Fisher-Senecor test was used to assess the adequacy of the regression equation with the test results. At the first stage of the analysis, the adequacy of the variance was determined, according to the following formula [35]:(5)$$S_{ad}^{2}=\frac{r\sum_{i=1}^{N}{\left(\bar{y_{i}}-\bar{\bar{y_{i}}}\right)}^{2}}{N-k-1}$$
where: $\bar{y_{i}}$ is the average value of measurement results in the *i*-th experiment, $\bar{\bar{y_{i}}}$ is the value calculated from the regression equation for the levels of input and output factors in *i*-th experiment, *k*—a number of terms in the regression equation (without a free term) after rejection of the insignificant terms, *N*–the total number of experiments.

Then, the value determined for the test coefficient *F*:(6)$$F=\frac{S_{ad}^{2}\left(y\right)}{S^{2}\left(y\right)}$$
was compared with the critical value determined from the Fisher-Snedecor distribution table.

In the FZ zone, the regression equation takes the following form:(7)$$\begin{array}{ll} \sigma =95.974+ & 143.825R_{t}+28.34R_{t}^{2}-1.15466R_{t}^{3}+13034.4R_{sk}+1606.52R_{t}R_{sk}\\ & +69.4897R_{t}^{2}R_{sk}+66511.5R_{sk}^{2}+3776.1R_{t}^{}R_{sk}^{2}+13.647R_{t}^{3}R_{sk}^{2}\\ & +93082.4R_{sk}^{3}+295.998R_{t}^{2}R_{sk}^{3} \end{array}$$

SP at a pressure of 0.4 MPa reduces the irregularity of the surface created during welding. The value of the Rt parameter decreases from the value of 21.32 to 11.36 (Figure 7b). Increasing the SP time to 4 min does not significantly change the value of the Rt parameter. The SP only causes deformations in the vicinity of the highest elevations of the profile, causing a change in the third order central moment (Figure 7a) from a positive value Rsk = 0.86 to a negative value (Rsk = −0.46 with SP time t = 2 min). Increasing the pressure to the value of 0.5 MPa causes a much greater smoothing of the surface profile. With the time t = 2 min, the SP only causes deformations in the area of the profile elevations, reducing the Rt parameter to 4.56.

Extending the SP time not only causes a deformation of the profile in the vicinity of the highest elevations, but also in the vicinity of the profile mean line, which results in an increase in the value of the skewness Rsk (from −0.35 to −0.17). The value of compressive stresses increases from the value of σ = −385 MPa to the value of σ = −413 Mpa (Figure 8). A further increase in the SP time results in a reduction in the skewness Rsk by 82% as a result of increasing deformations in the vicinity of the profile mean line. This causes the introduction of greater compressive stresses with σ = −470 MPa. Increasing the SP pressure to a value of 0.6 MPa enables the highest compressive stresses to be introduced. SP for a time t = 4 min allows the value of the Rt parameter of a value of 6.59 μm to be obtained, i.e., slightly lower than that obtained at the pressure of 0.5 MPa (Rt = 6.8 μm), but with a much lower value of the parameter Rsk (−0.21). This proves the impact of SP on the total height of the profile, which produces the highest value of compressive stresses σ = −609 MPa.

In the BM zone, the regression equation takes the following form:(8)$$\begin{array}{ll} \sigma =-787.635+ & 351.764R_{v}+70.839R_{v}^{2}-4.181R_{v}^{3}+28.324R_{t}-93.448R_{v}R_{t}\\ & +5.736R_{t}^{2}-0.486R_{t}^{3}-646.755R_{q}+60.9217R_{v}R_{q}+196.661R_{t}R_{q}\\ & -405.962R_{q}^{2}+23.820R_{q}^{3} \end{array}$$

Shot peening in the BM increased the height parameters of surface roughness compared to the surface of non-peened welded joints. The highest value of compressive stresses (*σ* = –358 MPa) was obtained at a pressure of 0.6 MPa and SP time *t* = 4 min. The highest value of compressive stresses is correlated with the highest value of the roughness parameter Rt and the smallest value of the Rq parameter (Figure 9a). Increasing the value of the Rv parameter also allows higher compressive stresses to be obtained (Figure 9b), especially when the Rq parameter is also reduced (Figure 9c).

## 4. Conclusions

The article presents the relationship between the surface roughness parameters and the stresses of the surface layer of welded joints subjected to the shot peening process in a mathematical form. This can be used in practice in the control of welded joints and in the selection of optimal parameters of the shot peening process, as it will allow for the reduction or avoidance of time-consuming and costly measurements of surface layer stresses. The tests of the SP process with variable pressure with a range of 0.4 to 0.6 MPa and time with a range of 2 to 4 min, carried out in accordance with the methodology of the PS/DK 3^2^ plan, showed that:The use of shot peening of TIG welded ASM 5504 stainless steel joints enables compressive stresses to be introduced in all the weld zones examined: FZ, HAZ and BM, thus contributing to an increase in the fatigue strength of the joint and, consequently, also to the reliability of the structure.The highest value of compressive stresses in both the FZ and BM was obtained at a pressure of 0.6 MPa and time *t* = 4 min. In the FZ, tensile stresses of *σ* = 127 MPa were recorded after the welding process. The SP process introduced compressive stresses in the FZ of *σ* = –609 MPa. In the BM, the initial value of compressive stresses was *σ* = –33 MPa, while after the SP process it increased to *σ* = –358 MPa.Analysis with the use of neural networks showed the relationship between the selected height parameters of the weld surface and the residual stresses. In the FZ, the most closely correlated parameters are Rt and Rsk. In the BM zone, the parameters Rv, Rt and Rq have the highest information capacity.As a result of the approximation, adequate regression equations were obtained for the most strongly correlated roughness parameters with residual stresses. Based on the equations developed, it was possible to estimate the level of compressive stresses in both the FZ and BM zones of the welded joint without the need to carry out laborious residual stress measurements. This allows the parameter settings of the SP process to be easily determined.As a result of shot peening in FZ, the roughness parameters Rt and Rsk correlated with stress parameters are reduced in the range of 46.84% to 69.09% and from 118.37% to 155.46% in relation to the base specimen, respectively.

## Figures and Tables

**Figure 1 materials-15-08835-f001:**
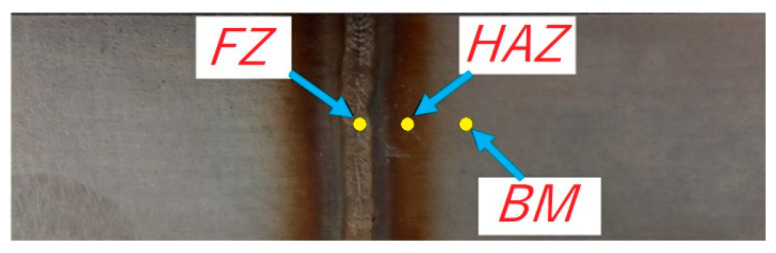
Zones for the measurement of residual stress.

**Figure 2 materials-15-08835-f002:**
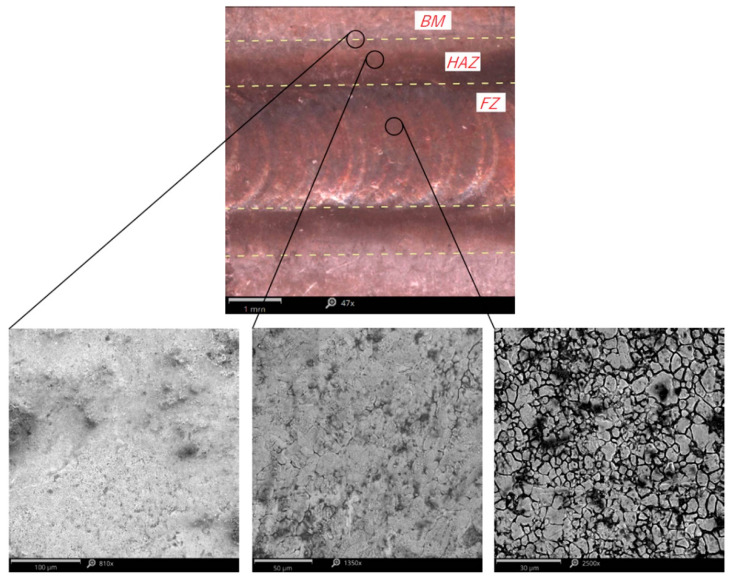
View of the weld face and SEM micrographs of the joint zones.

**Figure 3 materials-15-08835-f003:**
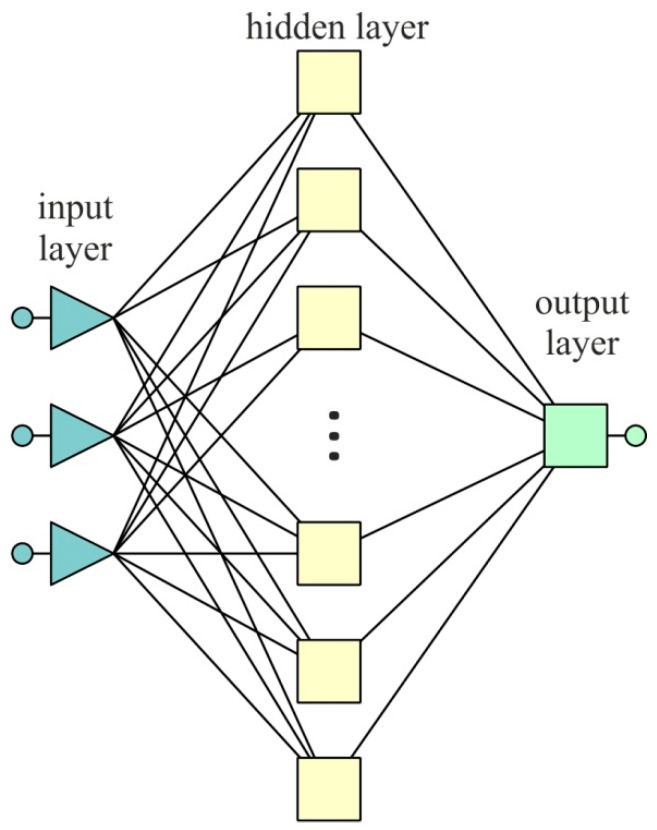
The architecture of the multilayer ANN.

**Figure 4 materials-15-08835-f004:**
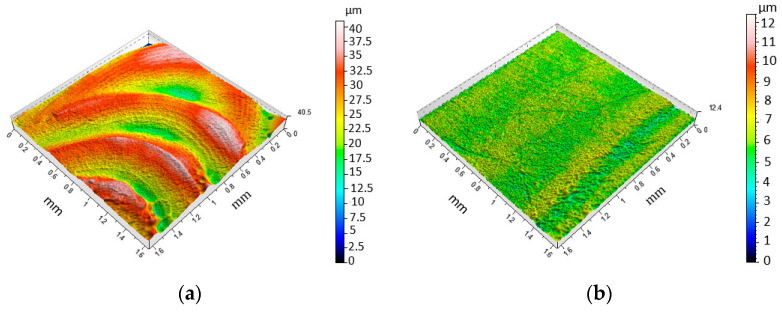
Three-dimensional roughness topography for variant 10 (specimen after TIG welding) in the (**a**) FZ and (**b**) BM.

**Figure 5 materials-15-08835-f005:**
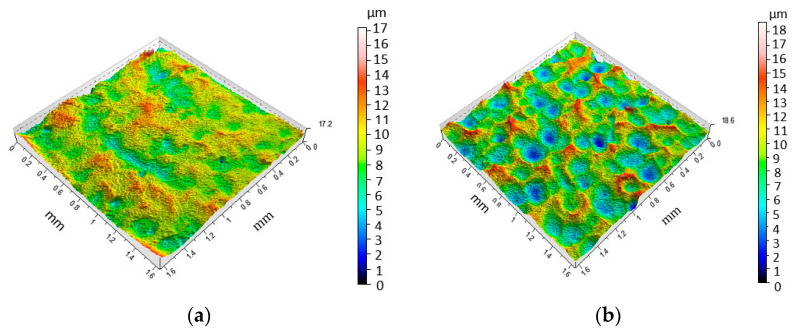
Three-dimensional roughness topography for variant 1 (specimen after shoot peening using the following parameters: *t* = 4 min, *p* = 0.6 MPa) in the (**a**) FZ and (**b**) BM.

**Figure 6 materials-15-08835-f006:**
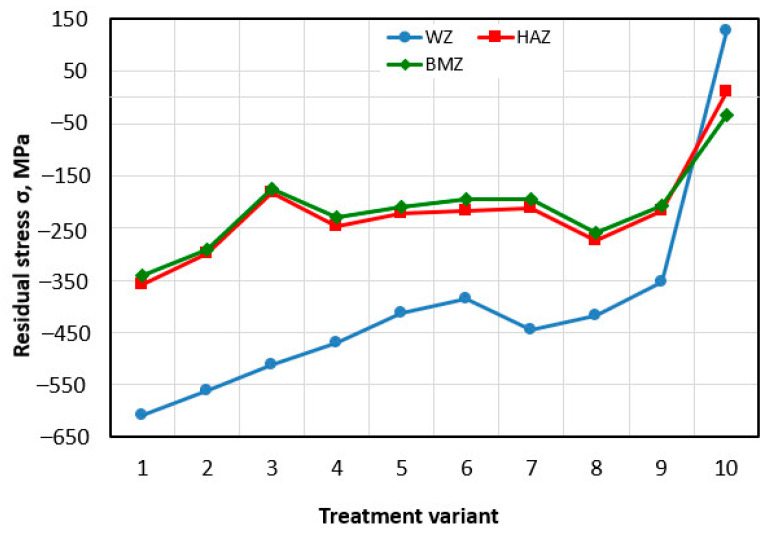
Effect of treatment variant on the value of residual stresses.

**Figure 7 materials-15-08835-f007:**
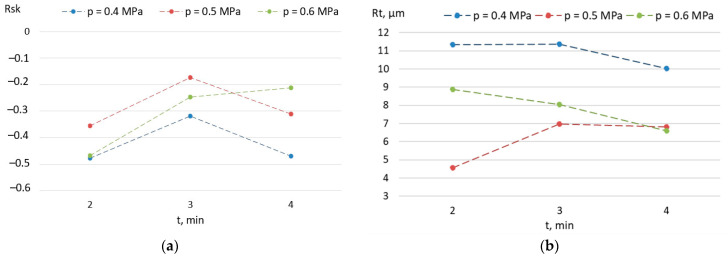
Effect of SP parameters on the value of the (**a**) Rsk and (**b**) Rt parameters.

**Figure 8 materials-15-08835-f008:**
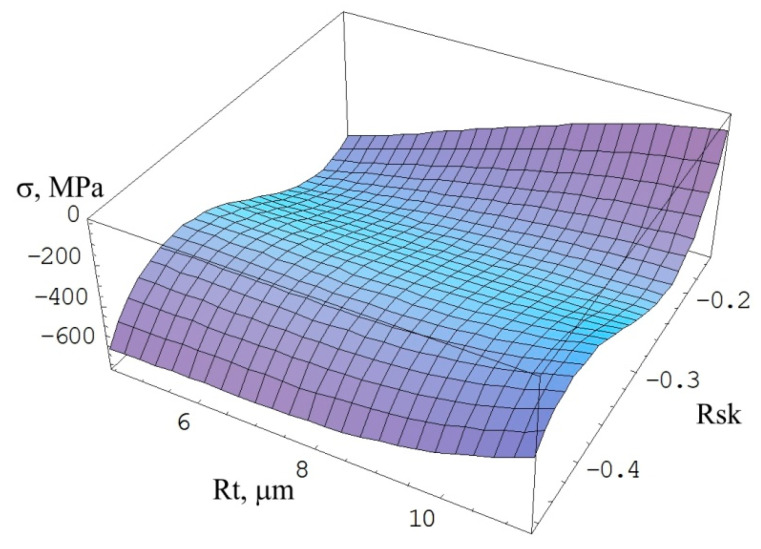
Effect of Rt and Rsk parameters on the values of compressive stresses.

**Figure 9 materials-15-08835-f009:**
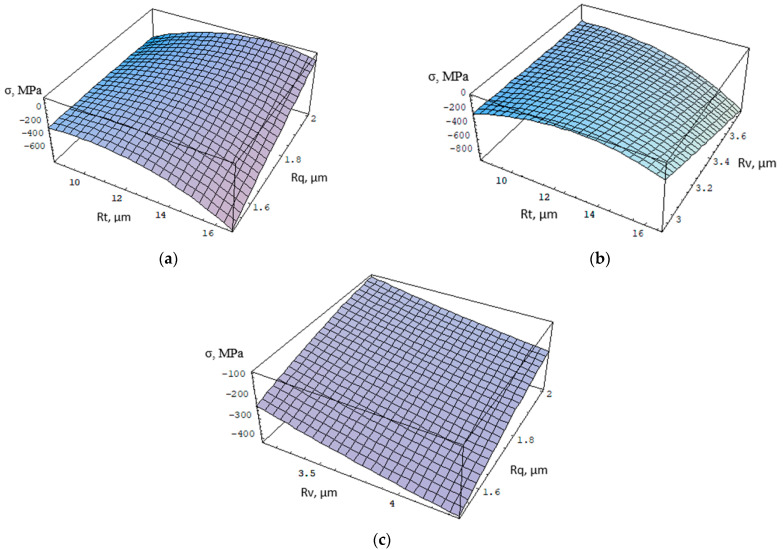
Approximation function for BM: (**a**) Rv = 3.10 μm, (**b**) for Rq = 1.74 μm, (**c**) Rt = 10 μm.

**Table 1 materials-15-08835-t001:** Chemical composition of AMS 5504 steel (%wt.) [29].

C	Si	Mn	P	S	Cr	Fe
0.15 max	1.0 max	1.0 max	0.040 max	0.030 max	11.5–13.5	Balance

**Table 2 materials-15-08835-t002:** Mechanical properties of AMS 5504 steel [29].

Tensile Stress Rm, MPa	Yield Stress Re, MPa	Elongation A, %
450 ÷ 510	205 ÷ 290	20 ÷ 34

**Table 3 materials-15-08835-t003:** Process parameters employed in the TIG welding of AMS 5504.

Process Parameters	Value	Unit
Welding speed	60	mm/min
Base current	15	A
Current pulse	40	A
Pulse current frequency	5	Hz
Shielding gas	Argon	-
Shielding gas flow rate	13	dm^3^/min
Tungsten electrode diameter	2.4	mm

**Table 4 materials-15-08835-t004:** Shot peening parameters of the experimental tests according to the PS/DK 3^2^ plan.

Variant No.	Pressure *p*, MPa	Time *t,* min
1	0.6	4
2	0.6	3
3	0.6	2
4	0.5	4
5	0.5	3
6	0.5	2
7	0.4	4
8	0.4	3
9	0.4	2
10 *	-	-

*—specimen not peened after TIG welding.

**Table 5 materials-15-08835-t005:** The G values for the roughness parameters and residual stress σ.

G Coefficient	Parameters							
	Ry	Rz	Rt	Ra	Rq	Rsk	Rku	σ
G_FZ_	0.2500	0.2224	0.2431	0.3195	0.2710	0.3137	0.2896	0.1580
G_BM_	0.3903	0.2739	0.3192	0.2598	0.4194	0.4363	0.3713	0.2049
G_HAZ_	-	-	-	-	-	-	-	0.2156

**Table 6 materials-15-08835-t006:** Surface roughness parameters in the FZ.

Variant No.	Rv, µm	Rz, µm	Rt, µm	Ra, µm	Rq, µm	Rsk	Rku
1	2.00	3.823	6.590	0.893	1.431	−0.212	2.437
2	2.79	5.130	8.037	0.971	1.207	−0.248	2.623
3	2.62	7.057	8.863	1.307	1.847	−0.469	2.483
4	2.50	4.913	6.803	0.960	1.173	−0.312	3.000
5	2.94	5.663	7.680	1.053	1.579	−0.158	2.430
6	2.29	4.380	6.983	0.932	1.133	−0.356	2.413
7	2.77	8.067	10.030	1.477	1.597	−0.469	2.353
8	2.31	6.213	11.363	0.901	1.480	−0.318	2.647
9	4.19	9.853	11.333	1.183	1.740	−0.477	2.977
10	27.690	40.882	21.320	4.000	5.045	0.860	3.820

**Table 7 materials-15-08835-t007:** Surface roughness parameters in the BM.

Variant No.	Rv, µm	Rz, µm	Rt, µm	Ra, µm	Rq, µm	Rsk	Rku
1	3.107	7.633	16.703	1.497	1.747	0.306	3.203
2	3.473	6.917	10.233	1.527	1.513	0.340	2.307
3	3.610	7.343	9.390	1.707	2.010	−0.191	2.060
4	3.580	7.063	9.220	1.337	1.633	−0.298	2.410
5	3.527	8.027	8.963	1.300	1.780	0.173	2.627
6	3.417	6.837	10.630	1.427	1.677	−0.288	2.253
7	2.917	6.197	8.813	1.270	1.513	0.038	2.243
8	4.050	7.223	9.553	1.370	1.687	−0.227	2.670
9	4.367	7.280	8.443	1.380	1.720	−0.349	2.607
10	5.968	12.400	3.047	0.377	0.484	−0.528	4.219

**Table 8 materials-15-08835-t008:** Regression statistic of the ANNs analyzed.

Parameter	7:7–7-1:1 (FZ)	7:7–9-1:1 (BM)
Data mean	−403.9	−213.5
Data S.D.	202.5709	81.4524
Error mean	−0.03575	−0.38136
Error S.D.	64.61516	14.33335
Absolute error mean	42.00425	11.16835
S.D. ratio	0.31897	0.17597
Correlation	0.94816	0.98440

**Table 9 materials-15-08835-t009:** Results of the sensitivity analysis for the network 7:7–7-1:1 (FZ).

Parameter	Ry	Rz	Rt	Ra	Rq	Rsk	Rku
Rank	3	6	1	5	4	2	7
Error	69.01206	64.8224	107.0941	65.2265	68.95482	104.9682	63.25987
Ratio	1.125821	1.057473	1.747068	1.064065	1.124887	1.712387	1.031983

**Table 10 materials-15-08835-t010:** Results of the sensitivity analysis for the network 7:7–9-1:1 (BM).

Parameter	Ry	Rz	Rt	Ra	Rq	Rsk	Rku
Rank	2	7	1	5	3	4	6
Error	51.42136	14.54843	55.47997	18.21497	46.83969	27.01425	14.64911
Ratio	1.125821	1.057473	1.747068	1.064065	1.124887	1.712387	1.031983

## Data Availability

The data presented in this study are available upon request from the corresponding author.

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
