# Peer review of "The Effect of Shot Peening on Residual Stress and Surface Roughness of AMS 5504 Stainless Steel Joints Welded Using the TIG Method"

_materials, 2022, doi:10.3390/ma15248835_

Round 1

Reviewer 1 Report

The paper presents the influence of various shot peening parameters on the residual stress distribution and surface roughness in three zones of AMS 5504 stainless steel joints welded. The quality of the manuscript is good, and it is suitable for publication on Materials after taking account by the following suggestions:

·         Reformulate the abstract in order to clearly show the strengths of this work.

·         Authors are advised to show the novelty of the work in the abstract as well as in the manuscript.

·         The choice of substrate, and TIG parameters must be justified.

·         Optical microscopy need to be added and discussed.

·         Did the plastic deformation phenomenon occurred during the SP process? the depth of the plastic deformation?

·         The discussion of the compressive stresses should be detailed.

·         What is the social contribution of your research? How your results and approaches are useful for industrial sectors?

Reviewer 2 Report

Manuscript ID Materials-2055835 entitled "The effect of shot peening on residual stress and surface roughness of AMS 5504 stainless steel joints welded using the TIG method" for journal of Materials has been reviewed.

- This article is comprehensive, logically organized and contains valuable information.

 However, there are few things need to be corrected and included in the manuscript for better understanding of carried research work to the readers.

+1- The novelty of the study should be further explained (in introduction…)

+2- In introduction section, more references should be added.  (especially about different studies)

+3- Tables 2, 3 and 8 should be checked.

+4- Figure 1, 2, 5, 8  and 9, the resolution should be increased. (and magnify)

+5- “In Materials and Methods” …. detail the processes further.

+6- Evaluation of SEM images should be increased.

+7- ….. The highest 238 compressive stresses can be observed after SP in the FZ ranging from −354 MPa (variant 239 9) to −609 MPa (variant 1), which is an increase of 379% and about 580% respectively in 240 comparison with the base variant (σ = 127 MPa). ………. More explain, Why?

+8-  ….. Hydrophilic CSA-LDH 203 film with surface hydroxyl groups generated during the sealing process further de-204 creased the contact angle to 4.6° (Figure 9c). However, after modification by PF, the 205 surface groups are changed to hydrophobic CF3(CF2)5(CH2)2- with low surface energy 206 [30] and…….. This section should be more detailed. (Specific explanations required)

+9- Conclusions section should be enriched.

+10- More literature studies should be added to the introduction and other sections (DOIs given below).

DOI-1   https://doi.org/10.1007/s13369-021-06243-w   (about different studies)

DOI-2  https://doi.org/10.26701/ems.989945  (about different studies)

DOI-2  https://doi.org/10.35193/bseufbd.1075980   (about different studies)

----------------------------------------------------

* It will be ready for publication after the specified corrections.

** I want to see article after the revision.

-----------------------------------------------------

Congratulations to the authors.

I wish the authors success in their future academic studies.

Kind regards.

Round 2

Reviewer 2 Report

Manuscript ID Materials-2055835 entitled “The effect of shot peening on residual stress and surface roughness of AMS 5504 stainless steel joints welded using the TIG method" for journal of Materials has been reviewed.

The authors have revised the manuscript carefully and the revised version could be published in the journal.

Decision- Accept

----------------------------------------------------------------------------

Congratulations to the authors.

I wish the authors success in their future academic studies.

Kind regards.